# Validation of Swarm Langmuir Probes by Incoherent Scatter Radars at High Latitudes

Hayden Fast [1], Alexander Koustov [1,*] and Robert Gillies [2]

1 Institute of Space and Atmospheric Studies, Department of Physics and Engineering Physics, University of Saskatchewan, Saskatoon, SK S7N 5E2, Canada
2 Department of Physics and Astronomy, University of Calgary, Calgary, AB T2N 1N4, Canada
* Correspondence: sasha.koustov@usask.ca

**Abstract:** Electron density measured at high latitudes by the Swarm satellites was compared with measurements by the incoherent scatter radars at Resolute Bay and Poker Flat. Overall, the ratio of Swarm-based electron density to that measured by the radars was about 0.5–0.6. Smaller ratios were observed at larger electron densities, usually during the daytime. At low electron densities less than $3 \times 10^{10} \, \text{m}^{-3}$, the ratios were typically above 1, indicating an overestimation effect. The overestimation effect was stronger at night and for Swarm B. It was more evident at lower solar activity when the electron densities in the topside ionosphere were lower.

**Keywords:** swarm satellites; incoherent scatter radars; electron density; topside ionosphere; polar cap; auroral zone

## 1. Introduction

In recent years, significant attention has been paid to the development of empirical models of the electron density distribution in the Earth's ionosphere, e.g., ref. [1–5]. There are practical needs behind these models, e.g., ref. [6], but there is also a general interest in understanding the physical processes behind plasma creation and its redistribution following strong solar activity events and slow changes occurring as the solar cycle progresses [7–9]. Efforts to improve the International Reference Ionosphere (IRI) family of comprehensive global-scale ionospheric models [10–12] and to develop models for specific ionospheric regions, e.g., ref. [1], have recently intensified. New comprehensive models have also been proposed, for example, the Empirical Canadian High Arctic Ionospheric Model [13,14].

Electron density profiles above the F region peak are of special interest for all empirical models. This region is difficult to investigate experimentally. It is not a surprise that several analytical and numerical approaches have been developed, and semi-empirical models have been proposed, e.g., ref. [5,15–17]. One important step in this area was the development of the Ne-Quick semi-empirical model [12,18–20], which became part of the IRI model [11].

Experimentally, one can study the topside ionosphere with ground-based incoherent scatter radar (ISR) [21–23]. Unfortunately, these radars operate for limited time periods and cover only the space near their zenith. Measurements obtained by satellites provide global coverage and over the years, significant data sets have been accumulated. Original topside sounding data analyses, e.g., ref. [24], have transitioned to studies with in-situ measurements using Langmuir probes (LP) [19,25–27]. In recent years, radio occultation (RO) measurements have become popular, e.g., ref. [20,28]. Measurements of Global Positioning System (GPS) signals from satellites on the ground also provide useful information on the electron density in the ionosphere, albeit indirectly for the topside, e.g., ref. [29].

Among other satellites with LP instruments onboard, a prominent role has been played by three Swarm satellites in operation since 2013 [30–32]. Their data can be used for

calibration and adjusting empirical and numerical/analytical models. The Swarm satellites fly at altitudes of 400–500 km and measure the electron density in situ with a temporal resolution of one second or better [31], i.e., a spatial resolution of less than 10 km. LPs are one kind of instrument on these satellites. The electron density can also be inferred from thermal ion imager (TII) instruments [32]. Data from the TII have only just started to be explored [33].

The LP method of electron density measurement is well established; however, each Swarm unit requires validation. Several recent publications [33–36] addressed this aspect of Swarm experimentation with LPs. It was found that, overall, the Swarm LPs report electron density ($N_e^{Swarm}$) values compatible with those measured on the ground and in space, although there is a tendency for LPs to underestimate the electron density by up to 30%. Data recalibration approaches have been suggested.

The Swarm validation work has been more extensive for middle and low latitudes and limited for high latitudes. Lomidze et al. [34] and Smirnov et al. [35] presented data for high latitudes obtained with the RO method. Although their results were in line with those reported for middle and low latitudes, there is a general concern about the quality of RO electron density measurements at high latitudes [37]. This is because the method assumes that the ionosphere is spherically layered, which can often be violated at high latitudes. Larson et al. [36] focused on Swarm validation at extreme high latitudes, in the polar cap, by comparing their data with measurements of the ISR radars operating at Resolute Bay (RB), Canada. The authors considered radar–satellite conjunctions with very close spatial coincidence. For this reason, their data set was limited. Performance of the Swarm LPs at the auroral zone latitudes has not been widely discussed.

The aim of the present study was to compare ISR–Swarm electron density measurements at auroral zone latitudes, thus expanding the previous ISR–Swarm work by Larson et al. [36].

## 2. Instruments

The Swarm mission incudes three satellites, A, B, and C, flying in near-polar low-Earth orbits with an orbital period of ~95 min [30–32]. In this study, we considered measurements after the satellites were positioned in such a way that Swarm A and Swarm C were separated by a time delay on the order of 7–10 s at altitudes of about 450 km with some longitudinal offset (100–200 km), while Swarm B, with an altitude of about 510 km, kept flying in a separate orbital plane. This configuration was achieved by May 2014.

Each Swarm satellite has a suite of instruments to measure magnetic and electric fields in space as well as the electron density and temperature. As mentioned, in this study, data from the LP instruments [31,32] were considered. A detailed description of LPs can be found in ref. [32,33,38]. Here, we mention that the Swarm LP sensors are traditional spheres sticking out of the satellite main body on 8 cm long posts. The spheres collect current under varying applied bias voltage. Instead of the traditional approach of building the volt–ampere (V–I) curve, the Swarm LP measurements are obtained at only three bias voltages: (a) at negative voltage so that the current collected by the probe is driven solely by ions, (b) at suitable positive voltage so that the electron current saturation is reached (linear part of V–I curve), and (c) at intermediate voltages in the middle of the V–I curve "knee" where the transition from the ion-driven current to the electron saturation curve occurs (retarded electron region). Because the Swarm database has so far been based on measurements in the ion saturation region, the data are often termed "ion densities" [33]. The ion density from the Swarm LPs is obtained under the assumption that the ionospheric ions at the Swarm heights are mostly $O^+$. The assumption is climatologically verified in the ionosphere at the Swarm heights so that we can safely consider the quasi-neutrality hypothesis and treat such an ion density as an electron density. We will use the term electron density.

There are two LP units on each satellite with low- and high-gains. This allows one to obtain reliable measurements of large electron/ion densities at low latitudes and much

lower densities in the polar cap. In the current work, only data from the high-gain units [37] were considered (ftp://swarm-diss.eo.esa.int/Level1b/Entire_mission_data/EFIx_LP (accessed on 1 July 2022).

This study compared the Swarm data with measurements of two ISRs, one in the polar cap at Resolute Bay (RB) and another in the auroral zone at Poker Flat (PF). Both radars are modular-type systems capable of quickly repositioning their beams and inferring the electron density in multiple range gates up to 600–700 km in height [39,40].

At RB, two identical ISR radars have been in operation: the North face and the Canada-oriented face [39,40]. The radars are located at a geographic latitude of ~74.7° and magnetic latitude of ~82.4°. In this study, only data from the RISR-Canada (RISR-C) were considered. In the present study, measurements in "imaging" mode were considered. The radar provides data in up to 51 beams. The second type of experiment, "world-day (WD) mode" with 11 beam positions, was not considered because these data were explored in ref. [36]. The advantage of the imaging mode is the ability to give electron density measurements in multiple beams so that an "average" profile can be obtained with local and strong deviations from a "typical" electron density profile being smoothed out. In addition, only data with 5 min integration times were considered to avoid occasional spurious data obtained with 1 min resolution. All RISR-C joint experiments with Swarm in 2016–2020 were considered. The electron density data used in this study were produced by the standard data processing procedure used for the RISRs, including calibration of each experiment with a co-located digital ionosonde (see details in ref. [36]).

The ISR radar at PF (geographic latitude ~65°; magnetic latitude ~66.1°), known as PFISR, is similar to the RISR radars in terms of hardware and the principle of data acquisition. Similar to RISR, the radar has multiple modes [41]. Measurements in four beams with 5 min resolution were considered in this study. The range resolution of PFISR measurements was ~50 km, providing altitude resolution on the order of 15–20 km. The PFISR radar, covering extended periods of observations in support of the "International Polar Year" project, has a much more comprehensive data set than the RISR-C radar and its data are a primary focus of the present work. Observations in 2014–2016 were selected for this study.

## 3. Methodology of the Comparison

The locations of the PFISR and RISR-C radars are shown in Figure 1. For PFISR, four pierce points at 450 km are shown by circled crosses; the middle point corresponds to the radar location. The PFISR electron density profiles were median-averaged over 4 beams. All measurements were considered irrespective of error values, and the "error" of measurements at each height was characterized by the standard deviation.

For RISR-C, the radar location is indicated by a yellow diamond. Measurements were obtained somewhat equatorward of its location. Pierce points at 450 km for 21 beams are shown in Figure 1 by circled crosses. For any of selected beams (out of the total available 51), the elevation angles were above ~55°; all other beams were judged to be too far away from the radar location. Originally, there were plans to conduct similar work with the RISR-North radar, but because the results obtained for the RISR-C radar were consistent and compatible with those from the previous comparison by Larson et al. [36], this work was left for future studies. RISR-C electron density profiles in multiple beams were median-averaged to obtain an electron density height profile characterizing a spatially large domain, similarly to the PFISR data handling.

The Swarm electron density values along their track were also median-averaged over points of measurements within $\pm 1°$ of the geographic latitude of the ISR radar location. The area of the Swarm data averaging was about 200 km in latitude, which was comparable to the region of the ISR data averaging. Longitudinally, a Swarm separation from the radar sites of up to 15° was allowed. The spatial regions where the Swarm measurements were classified as a "conjunction" with ISRs are shown by darker shading in Figure 1 for both PFISR and RISR-C. In terms of time, the Swarm data were considered if a satellite

was within the allowable area for comparison (dark sectors in Figure 1) any time within a 5 min interval of ISR measurement. Figure 1 shows the footprints for two Swarm satellite passes, one over each of the ISR regions of comparison. The Swarm satellites travel either equatorward (as shown over RB) or poleward (as shown over PF), and conjunctions for both cases were considered. To increase the statistics, the data from Swarm A and Swarm C were treated as separate points (for comparison with both ISRs), despite the fact that the temporal difference between the satellites was on the order of 10 s.

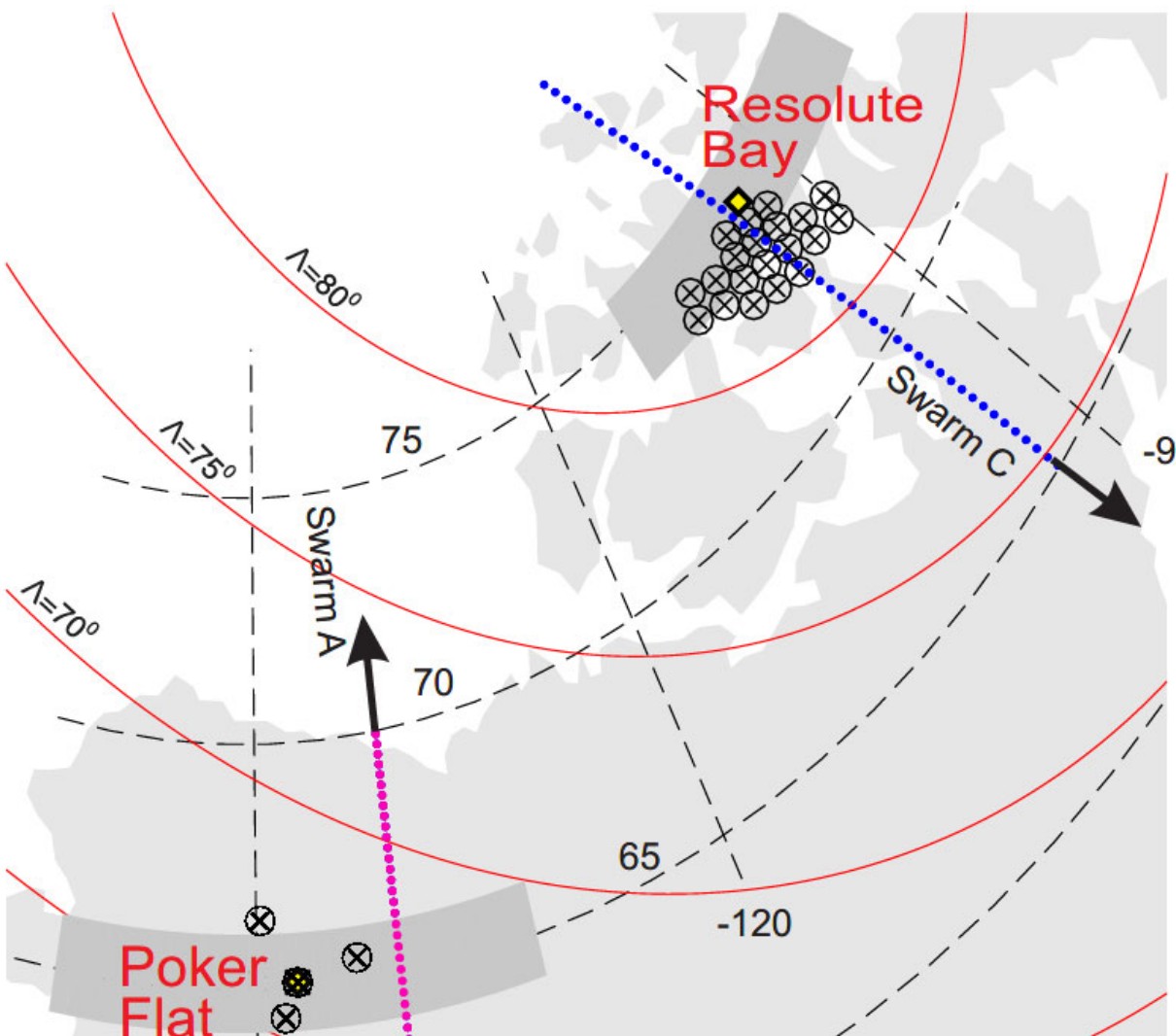

**Figure 1.** A map illustrating typical measurement coverage by RISR-C (Resolute Bay) and PFISR (Poker Flat) and the footprints of the Swarm satellites passing the area of ISR measurements in two different events. For the ISRs, an altitude of ~450 km was considered. Crosses indicate those locations for which an averaged electron density was computed and compared to Swarm data. Darker shaded segments around RB and PF outline the areas over which Swarm data were averaged along a trajectory. Shown by solid dots are Swarm C footprints for 27 July 2016 (around 16:35 UT, blue) and Swarm A footprints for 21 February 2015 (around 07:23 UT, pink). Arrows indicate the directions travelled by the satellites.

In the present study, the approach to ISR–Swarm data handling for conjunctions was more in line with the approach undertaken by others [33–35] and different from that adopted in ref. [36]. Larson et al. [36] considered RISR-C and RISR-North data with a 5 min integration time (in World Day mode), and the Swarm data were averaged over time periods of a conjunction that was defined as a satellite separation from the center of a radar

gate by less than 200 km in distance and 20 km in height (from the height of the Swarm). Up to 2.5 min separation in time was allowed. Their data handling can be classified as a quasi-instantaneous value comparison. One disadvantage of their approach is that data at the "conjunction" height were often missing or of a poor quality for a specified beam but reasonable in other beams. The approach taken in the present study alleviates that problem. Another advantage of the present approach is in diminishing the effect of extreme localized electron density enhancements that frequently occur at high latitudes. In the polar cap, in the RISR-C vicinity, these enhancements are polar cap patches, while in the auroral (subauroral) zone, in the PFISR vicinity, these enhancements can be related to patches or intense particle precipitation [42].

## 4. Data Coverage

Figure 2 provides information on data coverage for three comparisons, PFISR with Swarm AC, PFISR with Swarm B, and RISR-C with Swarm AC. We will call them PFISR-AC, PFISR-B, and RISR-AC comparisons, respectively. RISR-C comparison with Swarm B was not performed because of difficulties in getting reasonable data coverage as RISR-C electron density profiles at or above 510 km were very noisy. In addition, the PFISR-B comparison showed results consistent with the pervious study by Larson et al. [36].

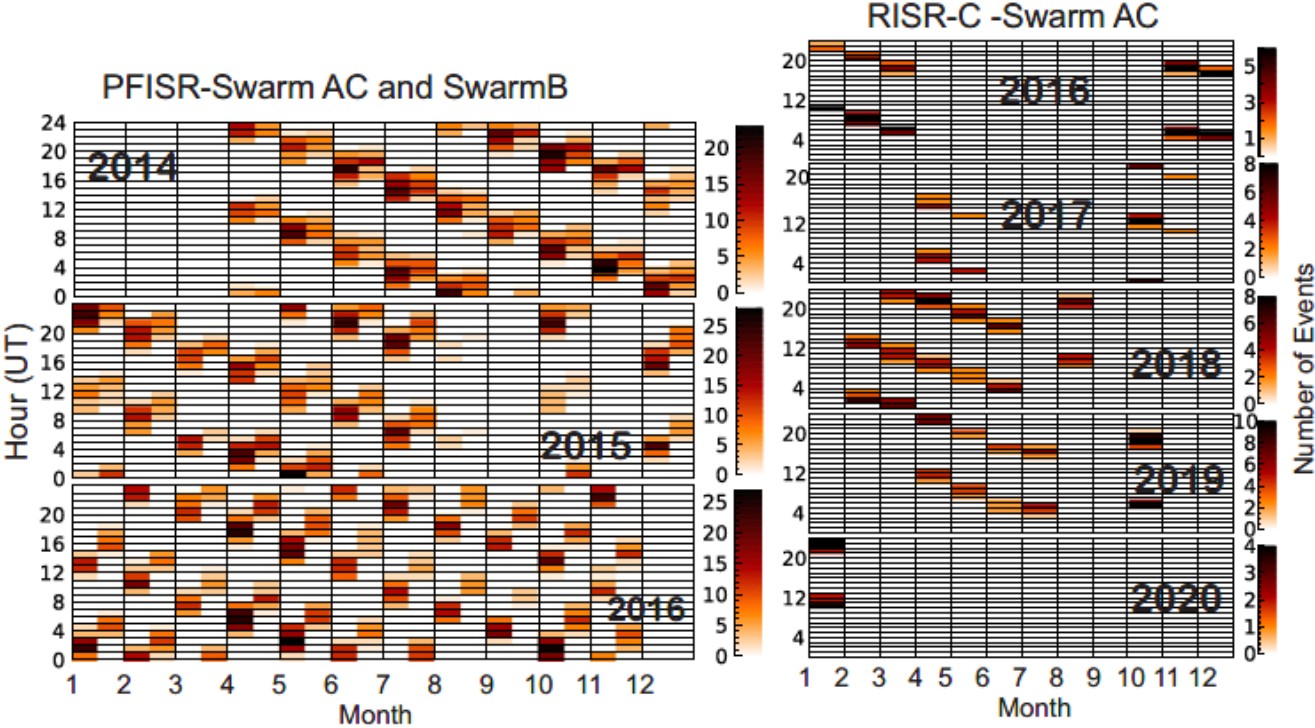

**Figure 2.** Number of joint ISR–Swarm measurements on the UT-month plane for observations over Poker Flat (PF), left panels, and Resolute Bay (RISR-C radar, Nunavut, Canada), right panels. For the Poker Flat data, each UT-month cell is divided into two halves with the left half for each month corresponding to the radar-Swarm AC conjunctions while the right half of each month corresponds to the radar-Swarm B conjunctions.

Figure 2 shows that for the PFISR-AC comparison, all months and UT sectors were covered cumulatively over 2.5 years, albeit not very uniformly, with some very limited gaps. The number of PFISR-B conjunctions was about two times smaller because for the PRISR-AC comparison, data from both satellites were combined into one data set.

Much poorer data coverage for the RISR-AC comparison is evident in Figure 2, particularly in the winter, despite having more years of RISR-C data. Total conjunction counts were about 2580 for PFISR-AC, 1260 for PFISR-B, and 420 for RISR-AC.

Figure 3 gives a sense of typical electron densities available for the obtained database in the form of cumulative histograms. The blue columns indicate that the PFISR-AC comparison had far more points of electron density measured by PFISR ($N_e^{PFISR}$) between 0 and $15 \times 10^{10}$ m$^{-3}$ and a gradual decrease in points at larger electron densities. For the PFISR-B and RISR-AC comparisons, low electron density data were more dominant. The median values of electron density were $14.2 \times 10^{10}$ m$^{-3}$, $8.8 \times 10^{10}$ m$^{-3}$, and $6.3 \times 10^{10}$ m$^{-3}$ for the PFISR-AC, PFISR-B, and RISR-AC comparisons, respectively. These numbers reflected changes in electron density with altitude (PFISR-AC versus PFISR-B) and latitude (PFISR-AC versus RISR-AC).

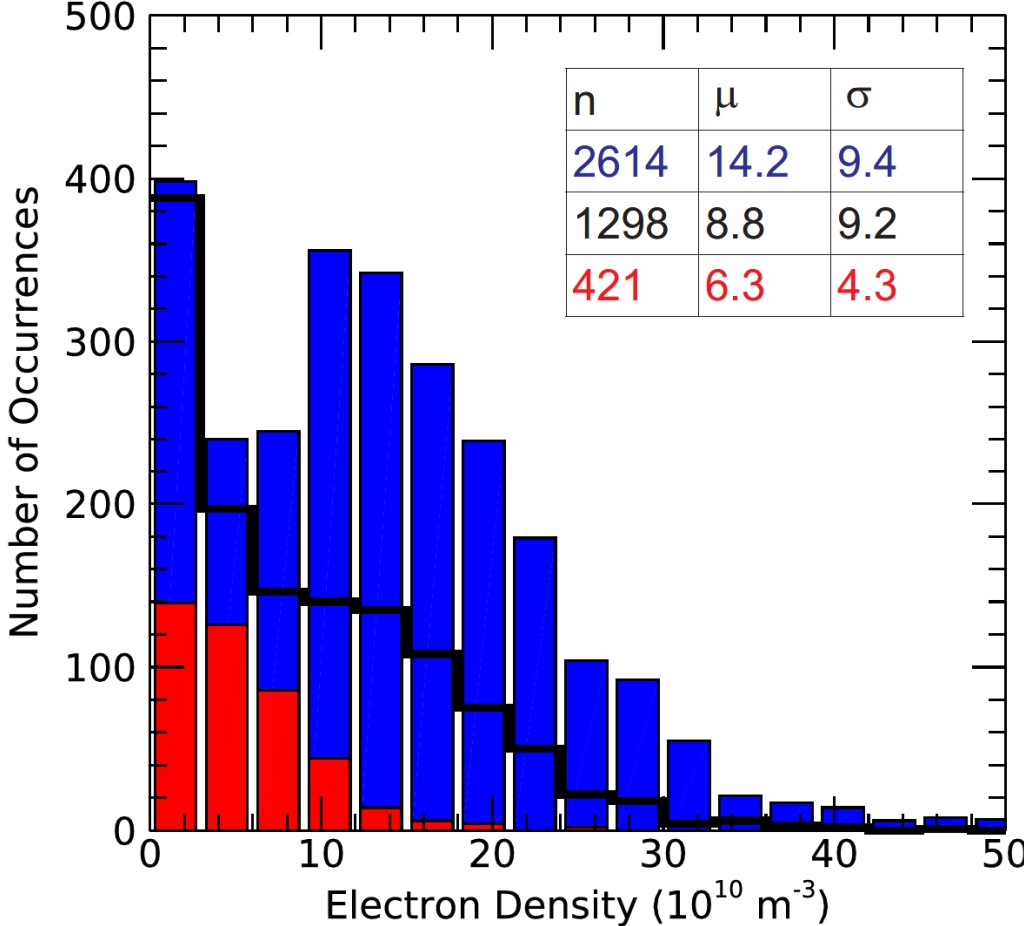

**Figure 3.** Histogram distribution for the occurrence of the electron density measured by the incoherent scatter radars PFISR and RISR-C for all conjunctions with Swarm satellites in 2014–2020. Blue columns are for the PFISR-Swarm AC conjunctions, red columns are for the RISR-C-Swarm AC conjunctions, and black columns are for the PFISR-Swarm B conjunctions. ISR electron density is plotted in units of $10^{10}$ m$^{-3}$. Total number of points n, median value $\mu$ (in units of $10^{10}$ m$^{-3}$), and standard deviation $\sigma$ (in units of $10^{10}$ m$^{-3}$) for each distribution are given in the upper right corner by respectively colored numbers.

## 5. Overall Scatter Plots

The first objective in assessing the Swarm data against the ISR measurements was to investigate whether the new approach of considering highly averaged data (in the present work) was compatible with the results by Larson et al. [36] comparing quasi-instantaneous electron density values at close locations. Figure 4a presents the data for the RISR-AC conjunctions in the form of a scatter plot.

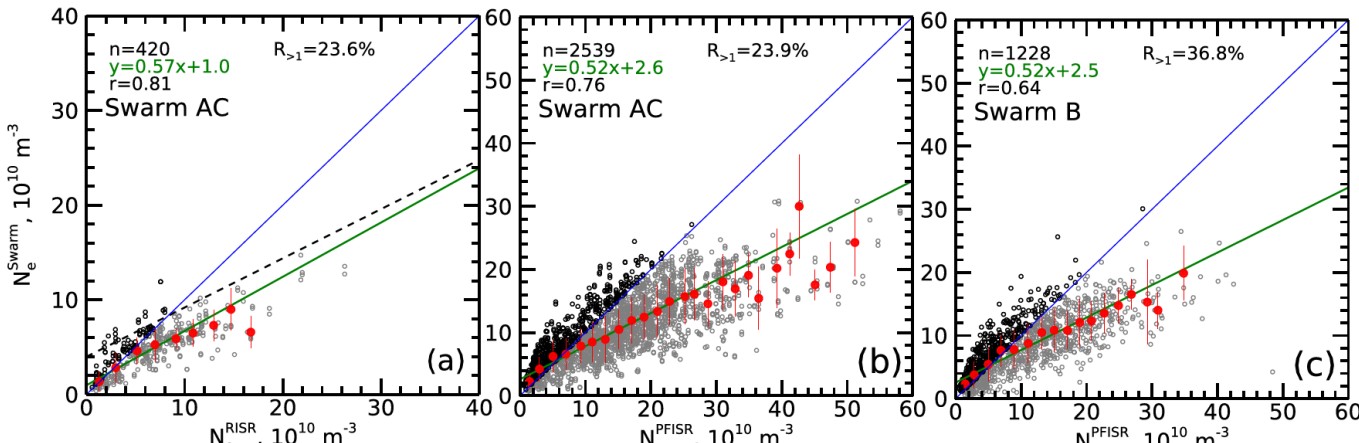

**Figure 4.** (**a**) Electron densities measured onboard the Swarm AC satellites versus electron densities measured by the Resolute Bay incoherent scatter radar RISR-C. Darker circles reflect measurements with $N_e^{Swarm}/N_e^{RISR} > 1$. Red solid dots are medians of $N_e^{Swarm}$ in bins of $N_e^{RISR}$ (size of the bin is $2 \times 10^{10}$ m$^{-3}$). Vertical red bars are the binned values of $N_e^{Swarm}$ +/− one standard deviation of $N_e^{Swarm}$. The green line is a linear fit line. The dashed line is a linear fit line reported by Larson et al. [36]. (**b**) The same as (**a**) but for the Swarm AC conjunctions with PFISR. (**c**) The same as (**b**) but for the Swarm B conjunctions with PFISR.

The points are scattered around the bisector of perfect agreement (blue line) with a trend of smaller $N_e^{Swarm}$ values at larger $N_e^{RISR}$ values. The tendency is obvious while looking at the red dots, representing medians of $N_e^{Swarm}$ in bins of $N_e^{RISR}$. The linear fit to the data, depicted by the green line, is somewhat different from the dashed line representing the linear fit line to a similar plot reported by Larson et al. [36], in their Figure 6. The slope of the line in Figure 4a indicates that, typically, the ratio was ~0.63, which was slightly larger than the values of 0.58–0.59 reported by Larson et al. [36]. This consistency implied that the comparisons in two ways were compatible. We note that because both instruments have uncertainties, least-squares fitting was performed in this study considering perpendicular offsets and minimizing the square of the perpendicular distances between the data and the best fit line. Larson et al. [36] used the same procedure.

The data shown in Figure 4a have one common feature with the previous work, namely, the presence of cases of Swarm overestimation (shown by darker color) at small $N_e^{RISR}$ values. The Swarm overestimations in Figure 4a seemed to be less significant in our case if one judges by the y-intercept of the fit line. It was close to 0 in Figure 4a versus about $4 \times 10^{10}$ m$^{-3}$ reported in ref. [36]. However, the number of cases of Swarm overestimation was larger here, occurring in about 24% of all points compared to about 12% in the data reported in ref. [36], in their Figure 5. We note that the total number of conjunctions in the present work was about half of that in ref. [36] but the number of crosses was about 4 times larger, since the study by Larson et al. [36] considered a beam-by-beam comparison and reported multiple conjunctions per passing.

Figure 4b,c show scatter plots for the PFISR-AC and PFISR-B comparisons in the same format as the RISR-AC comparison in Figure 4a. The number of available points was 6 times larger for the PFISR-AC comparison and 3 times larger for the PFISR-B comparison. The general tendencies here were the same as in Figure 4a and similar to the plots reported in ref. [36]. The linear fit line for the PFISR-AC comparison agreed well with the fit line reported by Larson et al. [36]. The number of Swarm overestimation cases was significantly larger for the PFISR-B data, with 37% versus 24% for the PFISR-AC data.

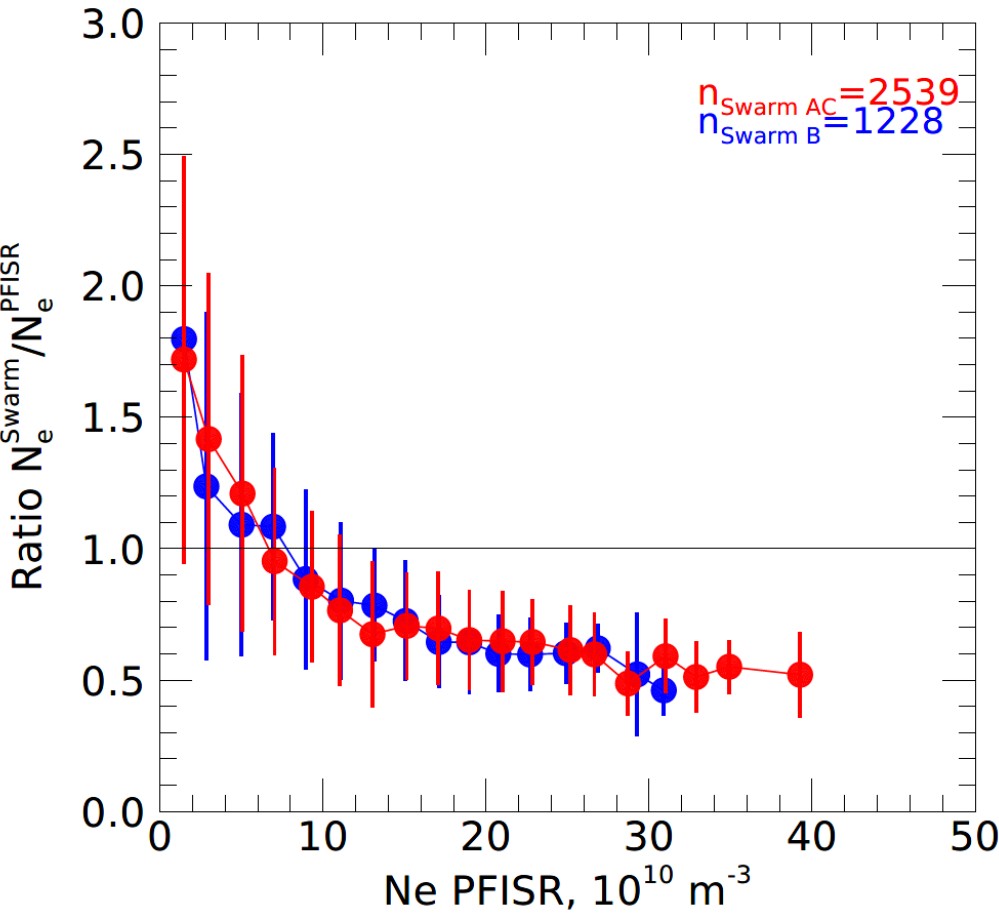

**Figure 5.** Medians of the ratio $R = N_e^{Swarm}/N_e^{PFISR}$ as a function of the electron density measured by PFISR, $N_e^{PFISR}$. The bins of $N_e^{PFISR}$ have a step of $2 \times 10^{10}$ m$^{-3}$. Vertical bars for each bin are the median value of the ratio $+/-$ one standard deviation of the ratio. Red and blue colors characterize data for PFISR conjunctions with Swarm AC and Swarm B, respectively.

To further investigate the overestimation effect, the ratio $R = N_e^{Swarm}/N_e^{PFISR}$ was plotted versus $N_e^{PFISR}$ binned in steps of $2 \times 10^{10}$ m$^{-3}$, as shown in Figure 5. One can see that the ratio $R$ steadily increased toward smaller $N_e^{PFISR}$. $R$ values were greater than 1 for $N_e^{PFISR} < 5 \times 10^{10}$ m$^{-3}$ and became greater than 1.3 at $N_e^{PFISR} < 3 \times 10^{10}$ m$^{-3}$. The tendencies were very similar for the PFISR-AC and PFISR-B comparisons, implying that the effect did not strongly depend on the height of the joint measurements.

The comparisons performed suggested that the Swarm electron density overestimation effect is a common feature of the Swarm LP instruments and is significant whenever electron densities in the ionosphere are low.

## 6. Solar Cycle Trend for the Ratio *R*

Xiong et al. [33] presented data indicating that the Swarm overestimation occurrence rate changed as the solar cycle progressed. Our data set was too limited to explore the effect in great detail but sufficient to identify trends at the decaying phase of the solar activity cycle 24.

Figure 6 shows a contour plot for the number of cases of ratio $R = N_e^{Swarm}/N_e^{PFISR}$ as a function of time between 2014 and 2016, with the red-pink color corresponding to the largest counts. Overlaying the contours is a black line (connecting white circles) representing the monthly median values of the F10.7 cm radio flux. In the text to follow, we will simply call the solar flux F10.7. The scale for the flux is on the right $Y$ axis.

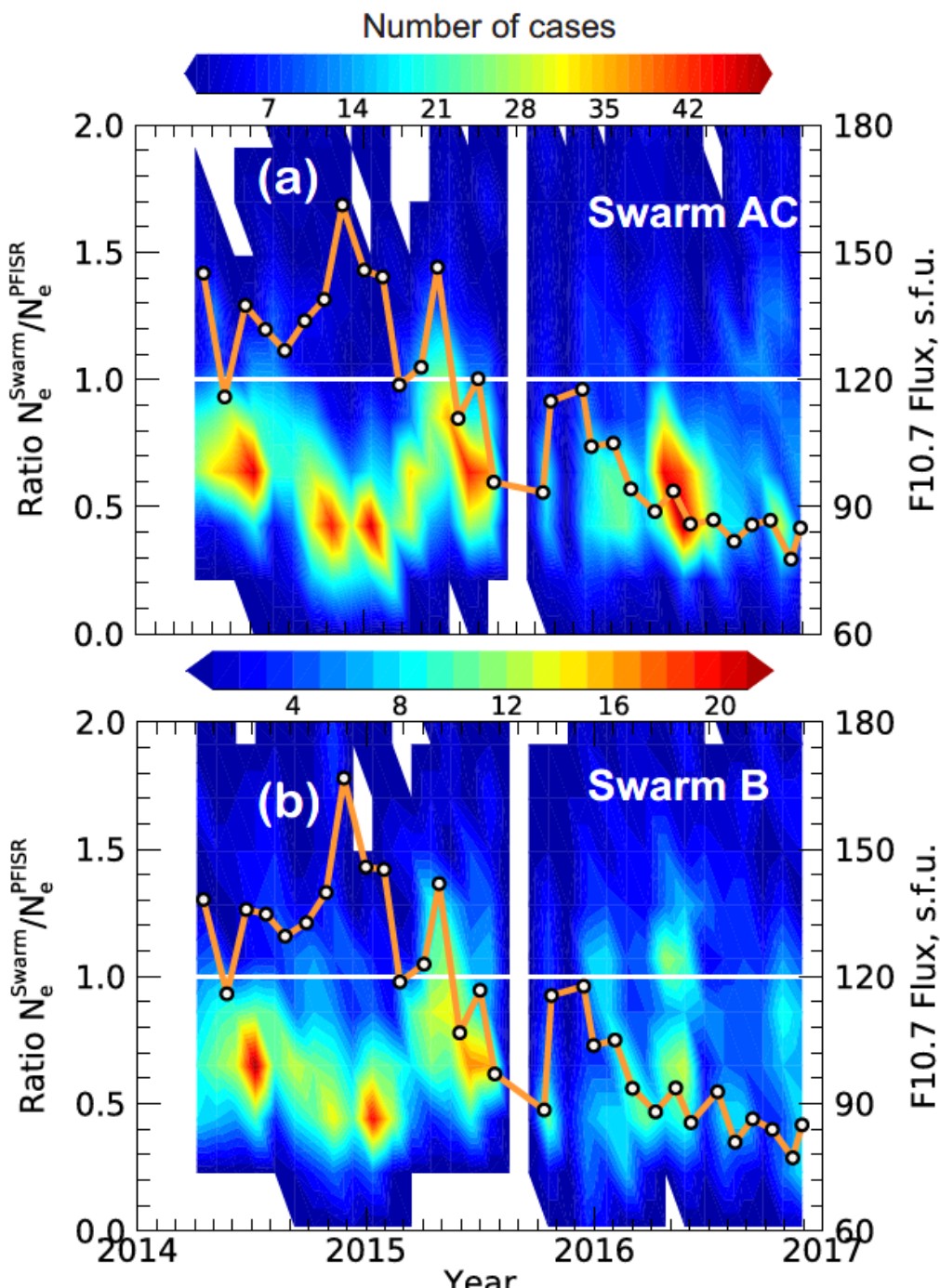

**Figure 6.** (**a**) A contour plot for the occurrence of the ratio $R = N_e^{Swarm}/N_e^{PFISR}$ for Swarm AC versus year of the radar–satellite conjunctions. The ratio is scaled according to the bar at the top of each plot. Black circles are the F10.5 cm radio flux (given in s.f.u. units). The scale for the radio flux is given on the right. (**b**) The same as (**a**) but for the Swarm B satellite.

A reduction in the F10.7 flux from about 120 s.f.u. in 2014 to about 80 s.f.u. by the end of 2016 was evident. The ratio *R* did not show an obvious trend with most cases being between 0.3 and 0.8 for both Swarm AC and Swarm B. One can claim that the distributions for *R* became flatter at lower solar activity, as the red-pink color became less present in 2016. This effect was more obvious in the Swarm B data (Figure 6b).

Upon closer examination of the plots in Figure 6, one can infer that the largest F10.7 values, occurring at the end of 2014 to the beginning of 2015, correlated with the

occurrence of the smallest *R* values. Additionally, a steady F10.7 increase from the middle of 2014 to the beginning of 2015 correlated with a steady decrease in *R*, and a short-lasting jump of F10.7 in the middle of 2015 was accompanied by a decrease in *R*. There was a short-lived enhancement of F10.7 at the beginning of 2016 that was not clearly accompanied by an increase in *R*, but the amount of data here was limited.

To investigate the solar cycle effect more quantitatively, median *R* values were plotted versus F10.7 in flux bins of 20 s.f.u., shown as solid dots in Figure 7. Here, a decrease in *R* with F10.7 was evident for both the PFISR-AC and PFISR-B comparisons. The decrease was not steady over the range of F10.7 values. Investigation showed that the largest solar flux values were observed at the end of 2014. During this period, the electron densities were somewhat depressed compared to what one would expect from the measured F10.7 values.

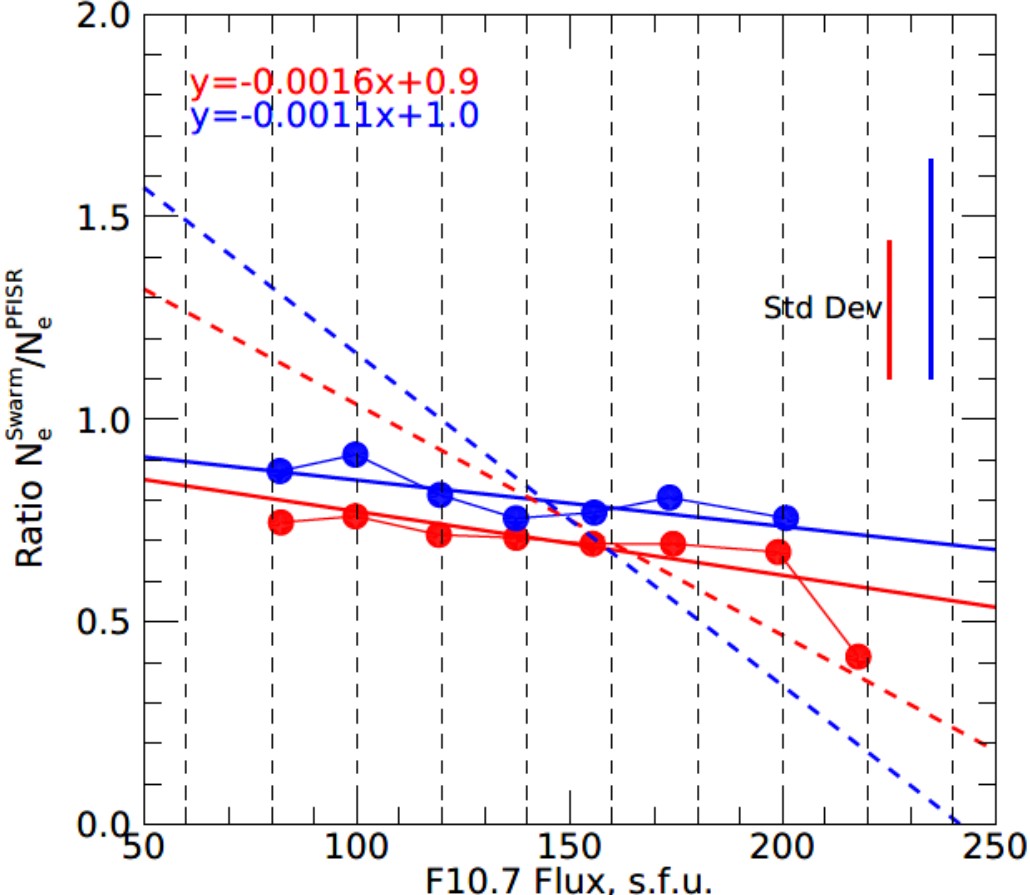

**Figure 7.** Medians of the ratio $R = N_e^{Swarm}/N_e^{PFISR}$ as a function of the F10.7 cm radio flux (given in s.f.u. units) for joint PFISR–Swarm measurements in 2014–2016. Red and blue colors characterize PFISR conjunctions with Swarm AC and Swarm B, respectively. Bins of the F10.7 cm radio flux have a step of 20 s.f.u. Typical standard deviations for *R* over all flux bins are shown by vertical lines, on the right. Sloped solid lines are linear fits, and parameters of the fit are given in the top left corner. Sloped dashed lines are the linear fit lines as reported by Xiong et al. [33].

The dashed-dotted lines in Figure 7 present the linear fit lines reported in ref. [33] for observations at low latitudes. The slopes of these lines were larger than those found in the present study, indicating a stronger dependence for observations at low magnetic latitudes.

## 7. UT/MLT Variations of Ratio *R*

Xiong et al. [33] presented data indicating that the Swarm overestimation effect was predominantly seen at night hours of MLT. We addressed this issue by plotting the PFISR–Swarm data as a function of UT, as shown in Figure 8.

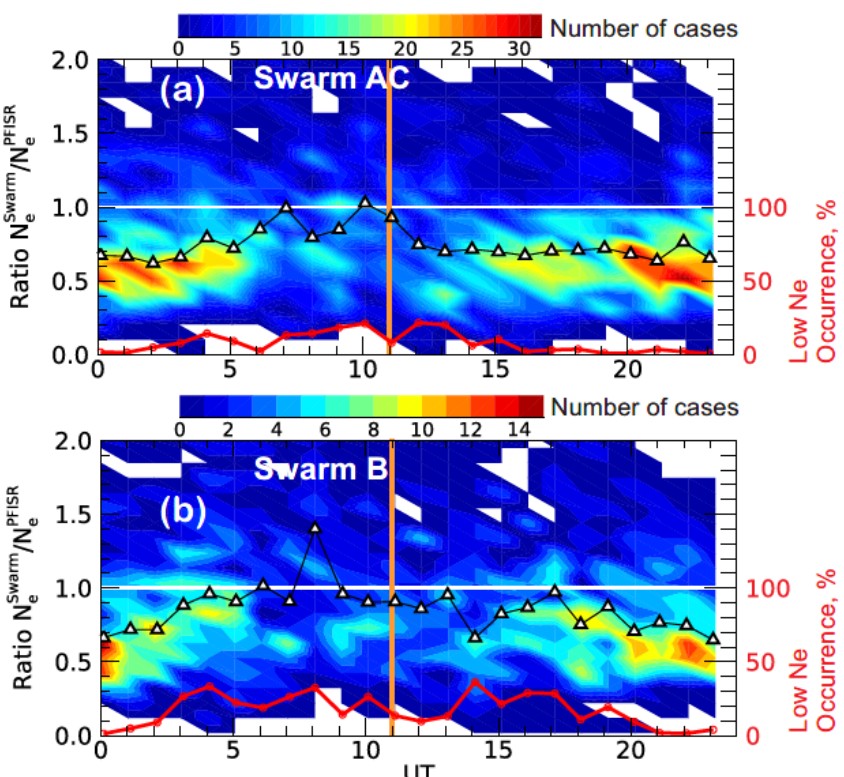

**Figure 8.** (**a**) A scatter plot for the occurrence of ratio $R = N_e^{Swarm}/N_e^{PFISR}$ for Swarm AC versus UT time of radar–satellite conjunctions. The scale for $R$ is represented by the bar at the top. The overlaid black-white triangles are hourly median $R$ values. The overlaid red line is the occurrence of low electron densities with $N_e^{PFISR} < 3 \times 10^{10}$ m$^{-3}$, given in percent of the total number of measurements for each one-hour bin. The vertical beige line denotes local midnight. The scale for the occurrence is shown on the right (*y*) axis. (**b**) The same as (**a**) but for the PFISR-Swarm B conjunctions.

Figure 8a,b show contour plots for the number of cases of $R$ as a function of UT time, binned over 1-hour intervals with a step in $R$ of 0.1. All collected data from 2014–2016 were considered. Overlaid on the scatter plots are medians of $R$ values over each 1-hour UT interval (black lines connecting white triangles) and median percentage of cases with $N_e^{PFISR} < 3 \times 10^{10}$ m$^{-3}$ (red line, the scale is on the right). The magnetic local midnight for Poker Flat is roughly 11:00 UT.

Figure 8a for the PFISR-AC comparison indicated that the nighttime distributions were flatter than in other time sectors, where strongly dominating $R$ values in the range of 0.5–0.8 (red color) were seen. The medians of $R$ were close to 1 during pre-midnight/midnight hours. This increase in $R$ values correlated with a general increase in the number of cases with small $N_e^{PFISR}$, which was consistent with the general trend reported in Figure 5.

From the data for the PFISR-B comparison presented in Figure 8b, one can make conclusions similar to those drawn from Figure 8a. The minor differences in Figure 8b were that $R$ values were close to 1 for a more extended period of 11 UT $+/-$ 6 h and the Swarm B overestimations during near midnight hours were stronger.

## 8. Discussion

Data reported in this study support the major conclusion of all previous publications that the Swarm LP instruments mostly underestimate the electron density, both at altitudes of about 450 km (Swarm AC) and about 510 km (Swarm B). Expanding the initial work by Larson et al. [36] where a "point-by-point comparison" was performed, we considered Swarm data averaged over a much larger space and involved ISR measurements over comparable regions (latitudinally) of the ionosphere. Our analysis of the new Swarm–RISR

data set for the RB area showed that the best linear fit lines had slopes anywhere between 0.4 and 0.6, which were somewhat smaller but compatible with those of Larson et al. [36].

The underestimation effect for the Swarm LPs was reported first in ref. [34] and then in ref. [35] and [33]. While the study by Lomidze et al. [34] characterized the scatter plots with a linear fit line with no y-offset, Smirnov et al. [35] and Xiong et al. [33] allowed for offsets and found them to be non-zero and positive, similar to ref. [36].

Xiong et al. [33] reported that the magnitude of underestimations changed with the solar cycle, being stronger at high solar activity (their Figure 6b,e), and had diurnal variation. Our data for the Resolute Bay area were too limited to make a definitive conclusion on the solar cycle effect because the RISR-C radar started operation in 2016 when the F10.7 flux was already low. We can say that the typical $R$ values were about the same in 2016–2020. Our analysis of the PFSIR and Swarm data for the Poker Flat area, which was based on a larger data set but spread over a limited period of time, confirmed the trend reported in ref. [33], albeit weaker for this location, as shown by Figures 6 and 7.

Attempting to investigate the effect further, we sorted the PFISR–Swarm data according to time sector of local solar time and assessed these limited data sets. The time sectors were introduced as follows: day (20-02 UT), dusk (02-08 UT), night (08-14 UT), and dawn (14-20 UT). The time slots were shifted by 1 hour to be more aligned with the magnetic local midnight at the PFISR location. First, histograms for the ratio $R = N_e^{Swarm}/N_e^{PFISR}$ were built, and the distribution medians and standard deviations were computed, similarly to ref. [36]. Second, linear fit lines for scatter plots of $N_e^{Swarm}$ versus $N_e^{PFISR}$ were produced for each data set.

The medians and standard deviations of $R = N_e^{Swarm}/N_e^{ISR}$ are given in Table 1, while information on the linear fit lines in the form $N_e^{Swarm} = a \cdot N_e^{ISR} + b$ is given in Table 2. In both Tables, either Swarm AC or Swarm B data were considered for the Swarms while either RISR-C or PFISR data were considered for the ISRs.

**Table 1.** The Median Value μ and the Standard Deviation σ for a Histogram Distribution of the Ratio $R = N_e^{Swarm}/N_e^{ISR}$ and Number of Points n or all Conjunctions between RISR-AC, PFISR-AC, and PFISR-B.

| Time Sector | | RISR-AC | PFISR-AC | PFISR-B |
|:---:|:---:|:---:|:---:|:---:|
| | μ | 0.73 | 0.67 | 0.70 |
| Day | σ | 0.44 | 0.36 | 0.46 |
| | n | 129 | 786 | 376 |
| | μ | 0.63 | 0.76 | 0.87 |
| Dusk | σ | 0.38 | 0.49 | 0.55 |
| | n | 86 | 655 | 317 |
| | μ | 0.78 | 0.85 | 0.98 |
| Night | σ | 0.36 | 0.53 | 0.61 |
| | n | 128 | 487 | 226 |
| | μ | 0.79 | 0.70 | 0.83 |
| Dawn | σ | 0.41 | 0.43 | 0.55 |
| | n | 77 | 611 | 309 |
| | μ | 0.74 | 0.72 | 0.83 |
| All | σ | 0.40 | 0.45 | 0.55 |
| | n | 420 | 2539 | 1228 |

Table 1 indicated that that the underestimation effect was stronger during the daytime and at dawn when the ratio $R$ medians were the smallest. Table 2 indicated that the underestimation effect was also stronger in the day and dawn sectors where the slopes of the fit line were slower and the y-offsets were larger. The stronger underestimation effect at daytime correlated with the rare occurrence of low electron densities during this time, as shown in Figure 8. The correlation coefficients given in Table 2 ranged from 0.53 to 0.83, which indicated that the correlation between the data sets was not always great. We note that the data presented by Xiong et al. [33], in their Figure 6c,f, showed

stronger underestimations at nighttime compared to daytime MLT hours, i.e., a different diurnal trend.

**Table 2.** Coefficients of a Linear Fit $N_e^{Swarm} = a \cdot N_e^{ISR} + b$, Total Number of Points and Correlation Coefficient to a Scatter Plot of Electron Density Measured by Swarm Versus Electron Density Measured by ISRs for all Conjunctions RISR-AC, PFISR-AC, and PFISR-B. Values of b are Given in Units of $10^{10}$ m$^{-3}$.

| Time Sector | | RISR-AC | PFISR-AC | PFISR-B |
|---|---|---|---|---|
| | a | 0.41 | 0.45 | 0.36 |
| Day | b | 2.98 | 3.96 | 4.75 |
| | n/r | 129/0.71 | 786/0.74 | 376/0.53 |
| | a | 0.50 | 0.57 | 0.71 |
| Dusk | b | 1.17 | 2.15 | 0.90 |
| | n/r | 86/0.70 | 655/0.77 | 317/0.80 |
| | a | 0.67 | 0.68 | 0.63 |
| Night | b | 0.49 | 1.21 | 2.19 |
| | n/r | 128/0.83 | 487/0.75 | 226/0.73 |
| | a | 0.56 | 0.53 | 0.49 |
| Dawn | b | 0.82 | 2.15 | 2.43 |
| | n/r | 77/0.76 | 611/0.74 | 309/0.57 |
| | a | 0.57 | 0.52 | 0.52 |
| All | b | 1.00 | 2.6 | 2.5 |
| | n/r | 420/0.81 | 2539/0.76 | 1228/0.64 |

Previous studies identified several possible reasons for the Swarm underestimations [33–36]. We briefly comment on two of them. Oyama and Hirao [43] suggested that surface contamination of LP electrodes can result in a decrease in the current through the probe, resulting in a lower measured electron density. One would expect the underestimation effect to be more pronounced at larger electron densities. The results of our study supported this hypothesis. However, if this effect is mostly responsible for the Swarm underestimations, it should be stronger at later periods of the mission with a general degradation of the instrument. Our analysis of data for the RB location (data not presented here) did not support this expectation, as typical *R* values were about the same in 2016 and 2020.

Smirnov et al. [35] and Xiong et al. [33] concluded that plasma "contamination" with light ions could seriously affect Swarm LP measurements. They cited the paper by Lira and Marchand [44], who considered a realistic model of Swarm LP probes and employed 3-D kinetic simulations to infer the current collected by the probe under typical conditions. They showed that the difference between the electron density inferred under pure O$^+$ plasma and plasma mixed with lighter ions can be on the order of 20%. We point out that some runs of their code showed a possibility of overestimations. In this view, the expected LP underestimation in simulations was correct only in a statistical sense. We note here that the ISR–Swarm comparison for the Poker Flat location showed results similar to those for the Resolute Bay location (Tables 1 and 2), and these results were compatible with the conclusions drawn from comparisons at middle and low latitudes [33–35] where potential sources/processes of ionization are not the same as at high latitudes. The similarity of the conclusions at various latitudes questions the dominant role of light ions in Swarm electron density underestimations.

Our data also confirmed the common occurrence of cases with Swarm LP electron density overestimations. Points with Swarm overestimations were evident in scatter plots reported previously [33–35], but many of them could be attributed to differences in the spatial and temporal resolutions of the instruments involved. For example, the uncertainty in the ISR radar calibration can be as large as 10%, as discussed in ref. [36]. To quantitatively evaluate the severity of the overestimation effect for various time sectors in our data sets,

we computed the percentage of points of Swarm overestimation with $R > 1.3$, as shown in Table 3.

**Table 3.** Percentages of Cases with Ratio $R = N_e^{Swarm}/N_e^{PFISR}$ more than 1.3 for Various Time Sectors.

| Time Sector | RISR-AC | PFISR-AC | PFISR-B |
|---|---|---|---|
| Day | 9.2 | 7.7 | 13.6 |
| Dusk | 5.8 | 13.6 | 20.5 |
| Night | 11.7 | 19.3 | 29.2 |
| Dawn | 13.0 | 9.3 | 18.5 |

Table 3 indicated that Swarm overestimations occurred more frequently at night/dawn for both Swarm AC and Swarm B, and they were more frequent for Swarm B at any time of a day. Qualitatively, these tendencies were recognizable in the plots of Figure 8.

Smirnov et al. [35] were the first authors to explicitly express the notion of Swarm overestimations. They reported the overestimation effect for the nighttime ionosphere at low–middle and equatorial latitudes (their Figure 7). No effect was evident in their plots at high latitudes, although the data statistics were lower there (their Figure 8). One needs to keep in mind that the radio occultation approach to electron density profile derivation (data used in ref. [35]) assumes a spherically layered ionosphere, which might not always be correct for the high-latitude ionosphere. Interestingly enough, the overestimation effect in ref. [35] was most evident for the electron densities measured by the Constellation Observing System for Meteorology, Ionosphere, and Climate (COSMIC) below $5 \times 10^{10}$ m$^{-3}$ (their Figure 7), which was consistent with our plot in Figure 5. It is worth noting that data by Lomidze et al. [34] at low electron densities, their Figure 4, showed rather agreement between the COSMIC and Swarm data.

Xiong et al. [33], comparing Swarm and Jicamarca ISR (near equatorial radar) data, showed that the Swarm overestimation effect varied with the solar cycle and MLT time of observations (their Figure 6). The effect was stronger in 2019–2020, the period of lowest solar activity. Our analysis for a high-latitude region based on a much larger data set (although covering a more limited period) showed an increase in Swarm overestimations with the decay of the solar activity, consistent with their results. Xiong et al. [33] proposed Swarm density correction equations based on values of the F10.7 cm radio flux. We comment that although their equations work well at near-equatorial latitudes and presumably for low-gain LPs, their applicability to other latitudes requires further investigation, as our comparisons for high-gain LPs showed much weaker trends, as shown in Figure 7.

We believe that Swarm electron density overestimations are not a random feature of the data. They occur very frequently during low electron densities in the ionosphere, as shown in Figure 5. Indirect supporting evidence is a systematic inference, shown in previous publications, that the linear fit line to Swarm electron density scatter plots versus data from an independent instrument has a non-zero and positive y-offset. Explaining the effect in terms of the physics or hardware operation features is a challenging task.

## 9. Conclusions

We summarize the results of this study as follows:

- Overall, electron densities measured by the Swarm satellites with high-gain LP instruments were smaller than those measured by the incoherent scatter radars at Resolute Bay in the polar cap and Poker Flat in the auroral zone. More extensive satellite–radar comparisons at Poker Flat showed that scatter plots of Swarm versus ISR electron densities had slopes of the best fit lines of ~0.52 for both Swarm AC and Swarm B. The y-offsets of the linear fit lines were ~$2 \times 10^{10}$ m$^{-3}$ and positive.
- The comparisons for the Poker Flat location showed that Swarm electron density underestimations occurred predominantly during the daytime, and the effect was stronger at higher solar activity. A stronger underestimation effect correlated with the occurrence of larger electron densities in the topside ionosphere.

- The comparisons for the Poker Flat location confirmed the existence of Swarm overestimations at high latitudes, earlier reported in ref. [33,35] for middle and low latitudes. Swarm overestimations were evident for low electron densities ($N_e^{PFISR} < 3 \times 10^{10}$ m$^{-3}$). The effect was observed in ~20% of conjunctions and more frequently for Swarm B.
- The Swarm overestimation effect was more frequent at lower solar activity, consistent with an overall decrease in the electron density in the topside ionosphere.

**Author Contributions:** Conceptualization, A.K.; methodology, H.F. and R.G.; software, H.F. and R.G.; validation, R.G. and H.F.; data curation, R.G.; writing—original draft preparation, H.F. and A.K.; writing—review and editing, R.G. All authors have read and agreed to the published version of the manuscript.

**Funding:** This research was supported by data analysis grant 424328 from the Canadian Space Agency and NSERC Discovery grant 411797. The University of Calgary RISR-C radar is funded by the Canada Foundation for Innovation and is a partnership with the US National Science Foundation and SRI International. The Poker Flat ISR is funded by the National Science Foundation, USA.

**Data Availability Statement:** ESA Swarm data are publicly available at ftp://swarm-diss.eo.esa.int/Level1b/Entire_mission_data/EFIx_LP (accessed on 30 August 2022). RISR-C and PFISR data are available at the Madrigal database http://madrigal.phys.ucalgary.ca (accessed on 1 August 2022) (both radars) or http://data.phys.ucalgary.ca (RISR-C) (accessed on 1 July 2022). The data used in this work are also freely available through the NSF-supported Open Madrigal Initiative (http://cedar.openmadrigal.org/openmadrigal (accessed on 1 August 2022).

**Acknowledgments:** Swarm is a European Space Agency mission with support from the Canadian Space Agency (CSA). The authors acknowledge the Swarm mission team efforts in collecting, processing, and distributing data for research and, particularly, for the extensive work on the Langmuir Probes data acquisition. Our thanks are also given to the SRI International team for making available data from the PFISR radar. We thank L. Lomidze and R. Marchand for valuable discussions of the results.

**Conflicts of Interest:** The authors declare no conflict of interest.

## Abbreviations

The following abbreviations are used in this manuscript:

| | |
|---|---|
| GPS | Global Positioning System |
| IRI | International reference ionosphere |
| ISR | Incoherent scatter radar |
| LP | Langmuir probe |
| PF | Poker Flat |
| PFISR | Poker Flat incoherent scatter radar |
| RB | Resolute Bay |
| RISR | Resolute incoherent scatter radar |
| RO | Radio occultation |

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
