# Peer review of "Validation of Swarm Langmuir Probes by Incoherent Scatter Radars at High Latitudes"

_remotesensing, doi:10.3390/rs15071846_

Round 1

Reviewer 1 Report

The study by Fast et al. presents validation of electron/ion densities by the Langmuir Probes (LPs) onboard the Swarm satellites by incoherent scatter radars at high latitudes. In particular, the authors use conjunction analysis to compare Swarm-LP data with the RISR and PFISR electron density observations. The findings are in good agreement with previous studies on Swarm-LP data quality and demonstrate that for low electron densities, there is an overestimation by Swarm-LPs (more frequently by Swarm-B), while for high electron densities there is an underestimation compared to the ISR data.

In my view, this study is well-designed and presented. The methodology is appropriate, and the conclusions are well supported by data. I think that this is a very valuable effort for validating the Swarm-LP observations, as no previous study had enough data points to look in detail at high latitudes. There are only a few things that I would like to ask the authors to clarify. Therefore, I think that the paper can be accepted after minor revision. The detailed comments are given below.

DETAILED COMMENTS:

(1) I would encourage the authors to rename the paper. I think that this is a really great work, but the current title does not do it justice. It misses a crucial part, that the comparison is specifically for high-latitudes which was not done before. Furthermore, the phrase "A contribution to", in my opinion, can be omitted and the meaning would be the same. I would suggest something along the lines of "Validation of Swarm electron densities by incoherent scatter radars at high latitudes". I think having a more precise title would increase the visibility and impact of the work.

(2) Line 35: "for limited periods" <- for limited time periods

(3) The paragraph at lines 51-56 says that LP data agree with other data sets and generally show underestimation. There is a slight contradiction with the rest of the paper, because your results, and also previous recent studies, suggest that sometimes the electron densities are overestimated and sometimes underestimated (based on geophysical conditions). Could you make this paragraph more aligned with the rest of the paper?

(4) General comment to Section 2 - it would be crucial to add which data version of Swarm LP was used in this study. Did you use baseline 05?

(5) Line 114 - the symbol for "degrees" is missing / turned into a question mark. Could you please fix it?

(6) Line 121 - "Observations in 2014-2016 were used for this study". Was there a reason not to include data after 2016, or is it simply for future work?

(7) Line 132- "55?" should probably be 55°(degrees). Same at line 143 ("15?")

(8) Lines 148-149: "Conjunctions occurring while the satellites were travelling either equatorward or poleward were considered". This is a bit unclear, because one is not sure how else they can travel?

(9) Figure 3 - the vertical axis should probably say "Number of occurrences". There is also an extra space in the caption after 10^10 m^-3 and before the bracket.

(10) General comment to section 7. I think it would be much easier for the readers to interpret the results if you converted UT to LT or MLT. (also in the discussion at lines 366-367).

(11) Line 469-470: "The effect was observed in 10-20% of conjunctions, more frequent for Swarm-B". I would say that this number is too low compared to Figure 4. It would probably be better to say "The effect was observed in >20% of conjunctions for Swarm-A and C and in >35% of conjunctions for Swarm-B".

Overall, this is a really great study and I think it will be a very valuable addition to the recent efforts on validation of Swarm LPs.

Author Response

Reply to Referee #1 comments:

We thank this reviewer for going through the paper and making valuable comments. Here is what we have done in the paper; all changes relevant to this reviewer comments are done in red

DETAILED COMMENTS:

(1) I would encourage the authors to rename the paper. I think that this is a really great work, but the current title does not do it justice. It misses a crucial part, that the comparison is specifically for high-latitudes which was not done before. Furthermore, the phrase "A contribution to", in my opinion, can be omitted and the meaning would be the same. I would suggest something along the lines of "Validation of Swarm electron densities by incoherent scatter radars at high latitudes". I think having a more precise title would increase the visibility and impact of the work.- Updated

(2) Line 35: "for limited periods" <- for limited time periods >>>> Corrected

(3) The paragraph at lines 51-56 says that LP data agree with other data sets and generally show underestimation. There is a slight contradiction with the rest of the paper, because your results, and also previous recent studies, suggest that sometimes the electron densities are overestimated and sometimes underestimated (based on geophysical conditions). Could you make this paragraph more aligned with the rest of the paper?

No, we cannot modify this paragraph because we describe here what has been reported by others. It is true that our conclusions are somewhat more mixed with respect to the underestimation effect. 

(4) General comment to Section 2 - it would be crucial to add which data version of Swarm LP was used in this study. Did you use baseline 05? >>> We added reference on FTP site where the data were taken from. It was and it is listed in section: Data Availability Statement

(5) Line 114 - the symbol for "degrees" is missing / turned into a question mark. Could you please fix it? Corrected

(6) Line 121 - "Observations in 2014-2016 were used for this study". Was there a reason not to include data after 2016, or is it simply for future work?

The main reason is that only 1-min data for > 2016 are available. We were reluctant to use them because the lead author conducted investigation of RISR-C data with 1-min and 5-min integration time and discovered that the data do not agree, consistently over the years, for the topside ionosphere while perfectly in agreement at lower heights. We do not know the exact reason for this effect.  We decided to avoid mixing up data sets of different quality. Data for > 2016 would not make much of a difference because the electron densities were generally very low, and 2016 data represent well observations at low solar activity and low electron densities.

(7) Line 132- "55?" should probably be 55°(degrees). Same at line 143 ("15?") >>Corrected

(8) Lines 148-149: "Conjunctions occurring while the satellites were travelling either equatorward or poleward were considered". This is a bit unclear, because one is not sure how else they can travel?

We are familiar with Swarm results where data for poleward and equatorward Satellite propagation show some differences. We wanted to stress that here we consider both directions of Swarm travel. We modified the statement.

(9) Figure 3 - the vertical axis should probably say "Number of occurrences". There is also an extra space in the caption after 10^10 m^-3 and before the bracket. 

Corrected both

(10) General comment to section 7. I think it would be much easier for the readers to interpret the results if you converted UT to LT or MLT. (also in the discussion at lines 366-367).

We prefer to stay with the UT time. The main reason is that the electron density in the ionosphere is mostly controlled by the Sun, i.e. UT time. Added a vertical line to figure 8 to denote local midnight, which is located near the center of the plot.  

(11) Line 469-470: "The effect was observed in 10-20% of conjunctions, more frequent for Swarm-B". I would say that this number is too low compared to Figure 4. It would probably be better to say "The effect was observed in >20% of conjunctions for Swarm-A and C and in >35% of conjunctions for Swarm-B".

Corrected to ~20%

Author Response

Reply to Referee #2 comments:

We thank this reviewer for studying our paper, noticing a number of errors, and making valuable comments. Here is what we have done in the paper; all changes relevant to this reviewer comments are done in blue.

Major comments

  • Lines 81-82: “Here we mention that the Swarm LP sensors are traditional spheres 81 sticking out of the satellite main body on long posts.

Not so long actually. The LP spheres are mounted on short poles 8 cm long (see e.g. Knudsen et al., JGR, 2017, your reference [32]) corrected

  • Line 84: “(a) at small voltage”

Not “small”, but “negative”. This is very important, otherwise you would not measure ion current. corrected

  • Lines 85-86: “(b) at relatively large voltage”

Better: “At suitable positive voltage so that the electron current saturation is reached”.  Corrected as recommended.

  • Lines 90-91: “We will use the term the electron density as the ionospheric plasma is known to be quasi-neutral with equal number of electrons and ions with high precision.”

Here there is a problem. What you say is certainly true in a plasma provided you are measuring the actual total ion density. This is not exactly the case of the Swarm’s Langmuir probes. In fact, one of the assumptions done for the calculation of Nion from the raw ion current measurement is that all the ions detected by the probes are O+. This is mostly, but not always true: especially at night time at low latitudes and during disturbed conditions at high latitudes you can have non-negligible penetration of H+ ions. So that it can happen that your measured ion current is not perfectly translated in a “true” ion density. This is mentioned in Section 3 of Catapano et al., 2022 (your reference [38]). I would therefore rephrase the sentence above in a more appropriate manner. For instance: “The ion density from Swarm LPs is obtained under the assumption that the ionospheric ions at the Swarm’s heights are mostly O+. The assumption is climatologically verified in the ionosphere at Swarm’s heights, so that we can safely consider the quasi-neutrality hypothesis and treat such an ion density as it was an electron density”. Corrected as recommended.

  • Line 101: as you did few lines below for PF radar, also for RB radar, for consistency, you should report in parentheses the geographic and geomagnetic coordinates location. Updated
  • Caption of Figure 4: (a), (b) and (c) labels are not represented in the actual panels. Either you write the letters in the panels or you use the notation “left/central/right panel” in the caption text. We increased the size of existed labels (a), (b) and (c)

  • Lines 245-246: “We note that the total number of conjunctions in the present work is about half of those in [36] but the number of crosses is about 4 times larger.”

What does that mean? What is the relevance of reporting the number of crossings if such crossings do not end up in good conjunctions?

Updated the statement.

This statement is to highlight the fact that the current paper covers more passes, more geophysically-different conditions. One has to keep in mind that Larson et al. (2021) had multiple points to report for one Swarm crossing (considered multiple ISR beams) while in this study one crossing gives one data point.

  • Line 257: “become greater than 2 at NePFISR < 3∙1010 m-3”.

This is not what I see in Figure 5. The maximum value reached by R is about 1.7. Agree, corrected to 1.3

  • Figure 6: the thin black line which represents the F10.7-cm radio flux from Sun is barely visible. I suggest to use a brighter color, such as yellow or orange. The lines were enhanced by applying different color and line thickness.
  • Line 273: “Figure 6 is a contour plot for the number of cases of ratio R” Please specify that R is RPFISR. >>> Introduced as recommended
  • Line 274: “between 2014-2017”
    It is rather 2014-2016, isn’t it? OK, corrected
  • Lines 282-284: “The other general conclusion from Figure 6 is that the Swarm B distributions tend to be somewhat more shifted towards the horizontal line of R=1.”

I do not see any evidence of this. I would remove this sentence. Agree, removed, one would need a hawk eye to see the effect.

  • Lines 289-291: “A short-lived enhancement of F10.7 at the beginning of 2016 is seen as a transition from a flat distribution of R values to more asymmetric distribution with domination of R~0.4.”

It is a bit hard for me to buy this speculation. Having a look to your Figure 2, one can see a remarkable lack of conjunctions, or at least a strong reduction, between August and November 2015. What you call a “flat distribution” could well be a statistical effect due to this reduction and I would be very careful to relate the “short-lived enhancement of F10.7 at the beginning of 2016” to the establishment of new features/structures in R distribution. I would either remove the sentence, or rephrase it by limiting yourself to describe what happens without speculations on correlations or transitions whatsoever.

We agree with this comment and suggestion. We modified the statement.

  • Lines 324-324: “The magnetic local midnight for Poker Flat is roughly 11:00 UT.” I would add a vertical line in Figure 8 to better mark the LT midnight. Added
  • Discussion, from line 364 on.

In the text and table captions you always refer only to PFSIR-Swarm data and R = NeSwarm/NePFISR, but in the tables also results for Swarm-RISR conjunctions are shown. Please add proper explanation in the text. Added

  • Table 2: the “correlation coefficient” between the two scatter plots of Swarm LP and ISR data is reported, but it is never discussed in the text. If authors think this has an added value, they should add few discussion lines in the text, otherwise it is not worth to be reported. We added several comments on correlation coefficients. We believe that there is value of them even without any written comments, as the reader can judge on the correlation.
  • Lines 400-413: Discussion on the contribution of lighter ions in the under/over estimation of density by Swarm.

I fear there could be a misunderstanding here. Xiong et al. (your reference [33]) first of all compare a LP dataset of ion density with the so called “Faceplate” (FP) dataset of ion density, always from Swarm, and then compared both with ISR data. They concluded that FP data are always statistically in better agreement with ISR with respect to LP data, especially because of a solar flux dependence of the LP dataset which seems not to occur for the FP data. Moreover, LP densities tend to an overestimation in the nightside wrt ISR, as also your results suggest, while FP densities do not. That LPs does not account for light ions is a fact, as I already mentioned in my first comment. On the other hand, density inferred from FP (which is, in fact, a kind of planar Langmuir probe) does not rely on any assumption concerning ion species. Pignalberi et al., Remote Sensing, 2022, https://doi.org/10.3390/rs14184679 , basing on results by Xiong et al., recalibrated the Swarm B LPs observations using the FP ones, and showed how the climatological distribution of the calibrated density pretty much improves in the nightside, getting closer to the distribution predicted by IRI model (their Figure 4). So, it is the overestimation of Swarm LP densities in the night hours that in large part depends on neglecting light ions. For the overall underestimation the reason could be different, and one can invoke other mechanisms.

This comment is off the topic of our paper. In our Discussion, we refer specifically to Swarm LP data, and ONLY to LP data, Figures 6 and 8 in Xiong et al. (2022). Faceplate data belong to a different instrument, these data never discussed in our paper and never meant to be thought of. We agree that the assumption of one ion specie is an issue for Swarm LPs, but bringing data from a different instrument into the Discussion would only confuse the reader. 

All sort of recalibrations of the original LP data are possibly OK, especially in a “climatological” sense, but these recalibrations do not resolve the actual problem – understanding the reasons for Swarm underestimations or overestimations. 

Typos

  • Line 19: “to the development the of”; Corrected
  • Lines 72-73: “flying in a near-polar low-Earth orbits”; Corrected
  • Lines 104-105: “The second type of experiment, “world-day (WD) mode” with 11 beam positions, were was not considered”; Strangely enough, our versions, both *.docx and *.pdf, do not show “was”. We do not know how to fix the journal version.
  • Line 114: the coordinates show a “question mark” (?) symbol instead of a “degree” symbol (°);
  • The reason for “typo” is that the journal software does not accept Mathtype fonts. We corrected to the plain version fonts.
  • Line 132: 55? >> 55°; Corrected “?
  • Line 135: semicolon after Larson et al. [36] to be replaced by a comma; Corrected
  • Line 143: 15? >> 15°; Corrected “?”
  • Lines 151-152: “despite the fact that the temporal difference between the satellites were was on the order of 10 s”; both our *.docx and *.pdf files, do not show “was”. We do not know how to fix the problem with conversion to *.pdf.
  • Line 154: “it is a different from”; corrected
  • Line 194: “is evident in Figure 2”; Corrected
  • Line 230: NeSwarR >> NeSwarm; Corrected
  • Line 233: “slightly larger than the values”; Corrected
  • Line 254: “the ratio R = NeSwarm/NePFISR is plotted”; Corrected
  • Line 257: NePFIS >> NePFISR; Corrected
  • Figure 6, Caption: NeSwarm/Knipfers >> NeSwarm/NePFISR;  Corrected
  • Line 318: “We address this issue with by plotting”; corrected
  • Line 353: “While the study of Lomidze et al.”; corrected
  • Lines 364-365: “we sorted the PFISR-Swarm data according to time sectors of local solar time”. corrected
